# am-ELO: A Stable Framework for Arena-based LLM Evaluation

**Zirui Liu** [1]  **Jiatong Li** [1]  **Yan Zhuang** [1]  **Qi Liu** * [1 2]  **Shuanghong Shen** [2]  **Jie Ouyang** [1]  **Mingyue Cheng** [1]
**Shijin Wang** [1 3]

## Abstract

Arena-based evaluation is a fundamental yet significant evaluation paradigm for modern AI models, especially large language models (LLMs). Existing framework based on ELO rating system suffers from the inevitable instability problem due to ranking inconsistency and the lack of attention to the varying abilities of annotators. In this paper, we introduce a novel stable arena framework to address these issues by enhancing the ELO Rating System. Specifically, we replace the iterative update method with a Maximum Likelihood Estimation (MLE) approach, m-ELO, and provide theoretical proof of the consistency and stability of the MLE approach for model ranking. Additionally, we proposed the am-ELO, which modify the Elo Rating's probability function to incorporate annotator abilities, enabling the simultaneous estimation of model scores and annotator reliability. Experiments demonstrate that this method ensures stability, proving that this framework offers a more robust, accurate, and stable evaluation method for LLMs.

## 1. Introduction

The rapid advancement of large language models (LLMs) (Jin et al., 2024b; Ouyang et al., 2025; Cheng et al., 2025) has led to the proliferation of "model arenas"—platforms designed to compare and evaluate multiple models, identifying their relative strengths and weaknesses (Chiang et al., 2024). These arenas play a critical role in driving innovation and shaping the deployment of cutting-edge LLMs across diverse applications. The ELO rating system (Elo, 1967), a well-established methodology for quantitatively

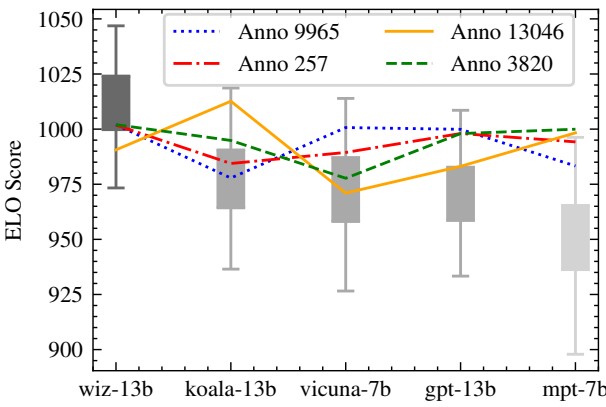

*Figure 1.* An example of ELO score. The error bar represents the standard deviation and the error line represents the difference between the maximum or minimum value and the mean value. The line chart represents the ELO scores estimated from the records of the specific annotator.

assessing the relative capabilities of competitors in games, forms the theoretical foundation for the evaluation systems in most existing model arenas (Bai et al., 2022; Boubdir et al., 2023).

A significant issue with the current ELO method is its instability, which can be attributed to two main factors: 1) From an algorithmic perspective, *the existing ELO method treats the data as dynamic, making the results highly sensitive to the order in which the data is presented* (Aldous, 2017; Li et al., 2024; Zhang et al., 2024a). In other words, when the same records are shuffled and re-evaluated, the ELO method often yields inconsistent scores. For instance, as shown in Figure 1, the significant error (highlighted in gray) complicates the comparison of models with similar abilities. 2) The judgment of human annotators varies across different aspects such as quality, relevance, and importance of texts. For example, in the line chart in Figure 1, different annotators provide inconsistent ELO scores for each model. However, the arena-based evaluation paradigm, which involves human participation, *overlooks these individual differences among humans* (Welinder & Perona, 2010; Raykar & Yu, 2011).

Ignoring this variability introduces biases and instability into the evaluation process, further undermining the credibility of both the results and the decisions derived from

[1]State Key Laboratory of Cognitive Intelligence, University of Science and Technology of China, Hefei, China [2]Institute of Artificial Intelligence, Hefei Comprehensive National Science Center, Hefei, China [3]iFLYTEK Co., Ltd, Hefei, China. Correspondence to: Qi Liu <qiliuql@ustc.edu.cn>.

*Proceedings of the 42nd International Conference on Machine Learning*, Vancouver, Canada. PMLR 267, 2025. Copyright 2025 by the author(s).

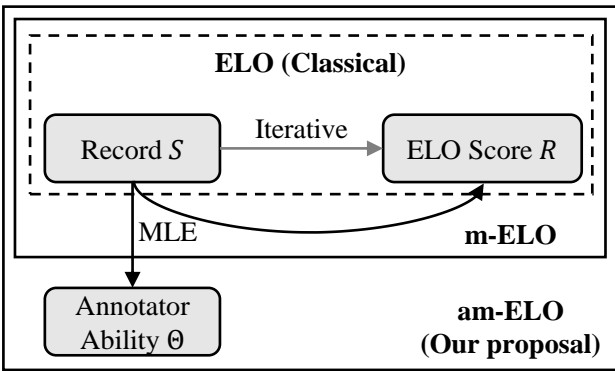

*Figure 2.* The traditional iterative ELO method and our proposed am-ELO method based on MLE.

them (Eickhoff, 2018). These instabilities diminish the interpretability and practical value of ELO scores, eroding confidence in the conclusions drawn from such evaluations, particularly when they are used to inform high-stakes decisions regarding model deployment or research directions.

In this work, we propose a novel stable arena framework to address these shortcomings. As illustrated in Figure 2, to mitigate the inconsistencies in ELO scores, we introduce a maximum likelihood estimation (MLE)-driven ELO rating method, referred to as m-ELO. By deriving the theoretical properties of this reformulation, we demonstrate that the proposed method produces consistent results without altering the fundamental principles of the original ELO method. Furthermore, to account for variability in annotator performance, we propose an annotator ability-aware enhancement method for ELO (am-ELO), grounded in psychometrics (Morizot et al., 2009; Furr, 2021). By modifying the ELO probability function, we estimate the annotator's ability and adjust their contribution accordingly, leading to a more accurate and equitable aggregation of evaluation results.

Through experiments on real-world datasets, we demonstrate that our framework effectively models annotators while ensuring the consistency of ELO scores. Furthermore, in simulation experiments, our method not only identifies anomalous annotators but also reduces the inconsistency of ELO scores to 30% compared to the traditional ELO method. This indicates that our approach effectively mitigates the instability inherent in the traditional ELO method.

## 2. Background and Related Work

Arena-based evaluation is an important subfield within the broader domain of LLM evaluation. Unlike traditional evaluation paradigms (Zellers et al., 2019; Hendrycks et al., 2020; Cobbe et al., 2021; Liang et al., 2022; Jin et al., 2024a), which typically assess a model's performance against predefined benchmarks, arena-based evaluation involves models competing directly with others. Current research in this area can generally be divided into three key categories: Battle Scenarios, Annotators, and Ranking Systems.

**Battle Scenario** The classic battle scenario is exemplified by the Chatbot Arena (Chiang et al., 2024), in which models respond to the same question and annotators compare their outputs. However, this approach are susceptible to the inherent biases of the annotators. To address this issue, several studies have incorporated multiple models working collaboratively to generate and evaluate responses, enabling iterative improvements (Zhao et al., 2024). Notable examples of this approach include LLMChain (Bouchiha et al., 2024) and ChatEval (Chan et al., 2024). While such strategies offer increased fairness, they come with trade-offs, including higher computational costs and potential instability.

**Annotator** In arena-based evaluation, the comparison of results typically involves human annotators (Cheng et al., 2024) or highly capable LLMs, such as GPT-4 (Achiam et al., 2023) and Claude (Anthropic). Additionally, some researchers have explored the use of specialized referee models for this task, such as PandaLM (Wang et al., 2023), JudgeLM (Zhu et al., 2023), and Auto-J (Li et al., 2023), which are designed to enhance the evaluation process.

**Ranking Systems for LLM Evaluation** Ranking systems play a crucial role in arena-based LLM evaluation (Busa-Fekete et al., 2014; Szörényi et al., 2015; Chernoff, 1992). Among the existing approaches, many arena-based methods rely on the ELO Rating System to model LLMs' capabilities (Coulom, 2007; Pelánek, 2016). The ELO rating system, grounded in the Bradley-Terry model (Hunter, 2004; Rao & Kupper, 1967), is widely used in competitive games (Sismanis, 2010; Ebtekar & Liu, 2021) to predict the likelihood of one competitor outperforming another based on their relative abilities. However, due to its dynamic nature, which is tailored for traditional competitive games, the ELO system introduces instability in LLM evaluation. To mitigate this instability, existing approaches typically perform multiple random shuffles of the annotated dataset and calculate ELO scores for each iteration (Sismanis, 2010). The statistical summary, such as the mean or variance of the scores across these shuffles, is then used as the final evaluation metric. Although this strategy provides a practical solution, it does not fundamentally resolve the inconsistency introduced by the sequential updates in the ELO method.

## 3. Preliminary

Arena-based Evaluation is a highly anticipated method in LLMs evaluation, where models are compared head-to-head on benchmarks or datasets and the results are annotated by evaluators. Let $S = \{(i, j, k, W_{ij}) | i, j \in [N], k \in [M]\}$

**Algorithm 1** The Traditional ELO Rating System

**Input:** Dataset $S$, Scaling Factor $K$, Init Score $R_{init}$
**Initialize:** Set of Scores $RS_i \leftarrow \emptyset$, Score of Models $R_i \leftarrow R_{init}$
Calculate ELO Score:
**for** $(i, j, W_{ij}) \in S$ **do**
    $R'_i \leftarrow R_i + K \cdot (W_{ij} - P(R_i, R_j))$
    $R'_j \leftarrow R_j + K \cdot (W_{ji} - P(R_j, R_i))$
**end for**
**Output:** ELO Score $(R_1, \cdots, R_N)$.

---

represent the comparative dataset we have collected, where $N$ is the number of models and $M$ is the number of annotators. Each element $(i, j, k, W_{ij}) \in S$ indicates that model $i$ and model $j$ engaged in a battle, and annotator $k$ provided the result $W_{ij}$. Specifically, $W_{ij} = 1$ indicates that model $i$ won the battle, $W_{ij} = 0$ indicates that model $j$ won, and $W_{ij} = 0.5$ indicates a tie. The goal of the arena-based evaluation is to estimate the ranking scores $R = (R_1, \ldots, R_N)$ for the models based on the record $S$.

**ELO Rating System**   The ELO rating system is a widely used method for ranking competitors based on pairwise comparisons. In the ELO system, each competitor (or model) is assigned a rating $R$, which represents its relative strength. When two models, $i$ and $j$, compete, their respective ratings, $R_i$ and $R_j$, are used to calculate the expected probability of each outcome: $P(R_i, R_j) = P(W_{ij} = 1) = \frac{1}{1+e^{-C(R_i-R_j)}}$, where $C$ is a constant that scales the difference in ratings. After observing the actual outcome of the match, the ratings are updated as follows:

$$R'_i = R_i + K \cdot (W_{ij} - P(R_i, R_j)),$$
$$R'_j = R_j + K \cdot (W_{ji} - P(R_j, R_i)), \qquad (1)$$

where $K$ is a scaling factor that controls the magnitude of rating changes. The pseudo-code for this process is shown in Algorithm 1. However, the existing ELO method is iterative and highly sensitive to the order of the data. This is irrational for LLMs' evaluation because evaluation can be seen as a static process (Zhan et al., 2024). Specifically, the errors introduced by the ELO method arise from the algorithm's dynamics rather than the data itself, which undermines the statistical significance of the ELO scores for many models.

Moreover, current algorithms do not account for differences in annotator abilities. They treat all annotators as if they have the same ability $C$, mixing annotation records randomly. This assumption can introduce bias and instability into the evaluation process.

# 4. Better Performance estimation with ELO

Earlier, we introduced the traditional ELO method and highlighted its key challenges, including ranking inconsistencies and the lack of consideration for annotator variability. To address these issues, this section presents a stable arena framework with improvements to the ELO method.

### 4.1. MLE for ELO (m-ELO) Estimation

The traditional ELO rating estimation method is based on an iterative algorithm, and the results are highly dependent on the order of the samples. This explains why ELO ratings often lack consistency. Inspired by the insensitivity of maximum likelihood estimation (MLE) to the sample order, we propose an MLE-driven ELO estimation algorithm, termed m-ELO. Specifically, for the record dataset $S$, its log-likelihood function can be expressed as follows:

$$\ln L = \sum_{(i,j,W_{ij}) \in S} W_{ij} \ln P(R_i, R_j) + W_{ji} \ln P(R_j, R_i),$$
$$(2)$$

where $P(R_i, R_j) = \frac{1}{1+e^{-C(R_i-R_j)}}$. The result of the MLE method, $(R_1^*, R_2^*, \ldots, R_N^*)$, can be obtained by solving the extreme point of the log-likelihood function using gradient descent. Specifically, for any given model $n \in [N]$, the gradient of the log-likelihood function with respect to its rating $R_n$ is:

$$\frac{\partial \ln L}{\partial R_n} = \sum_{(n,j,W_{nj}) \in S} C(W_{nj} - P(R_n, R_j)), \quad (3)$$

By comparing Equations 1 and 3, we observe that the two formulas share a consistent structure. This highlights the essence of the ELO algorithm: it performs gradient descent with a learning rate of $\frac{K}{C}$ on the MLE for each annotated sample. Gradient descent based on individual samples rarely converges, which reveals a key shortcoming of the traditional ELO method.

**Convergence Analysis**   Although the estimation results of the MLE method are not influenced by the sample order, another important consideration is whether the log-likelihood function has only one extreme point. If multiple extreme points exist, it could still lead to inconsistencies in the ELO rankings. Unfortunately, because ELO scores are relative, it is clear that if $(R_1^*, R_2^*, \ldots, R_N^*)$ is an extreme point, then $(R_1^* + \epsilon, R_2^* + \epsilon, \cdots, R_N^* + \epsilon)$ is also an extreme point. Thus, the extreme points of the log-likelihood function are not unique. However, when we fix the score of one of the models, we obtain the following theorem (Zermelo, 1929):

**Theorem 4.1.** *Assume $R_0 = 0$ and $|S|$ is sufficiently large, then the log-likelihood function $\ln L$ with respect to $(R_2, \cdots, R_N)$ is a concave function and has at most one extreme point.*

Drawing from Theorem 4.1, we can assert that the ELO score obtained through the MLE method is relatively stable between models, meaning that the difference in ability between any two models remains stable.

Replacing the iterative method with the MLE approach makes the ELO method more flexible. Additionally, it allows us to model annotator abilities during the evaluation process. In the next section, we will adopt ideas from psychometrics to propose a feasible modeling approach and analyze its interpretability.

## 4.2. Annotator Modeling m-ELO (am-ELO) Estimation

Although ability modeling is not commonly seen in LLM evaluation, many ability modeling methods have been developed in education and psychometrics (Liu et al., 2021; Wang et al., 2022; Zhang et al., 2024b; Zhuang et al., 2022; Liu et al., 2024). One prominent method is Item Response Theory (IRT) (Embretson & Reise, 2013; Zhu et al., 2022; Nguyen & Zhang, 2022; Polo et al., 2024). IRT posits that an examinee's performance on a test depends solely on its ability $\theta$ and the properties of the questions. The standard model is the two-parameter logistic (2PL) model, defined as: $P_j(\theta) = P(y_j = 1) = \frac{1}{1+e^{-a_j(\theta-b_j)}}$, where $y_j = 1$ indicates a correct response to question $j$, and $a_j$ and $b_j \in \mathbb{R}$ represent the discrimination and difficulty of question $j$.

As noted, the parameter $a$ in IRT can be interpreted as the discrimination parameter. Similarly, in the ELO method, the fixed value $C$ can also be understood as the discrimination parameter. To account for annotator variability, we replace the fixed value $C$ in the probability density estimation with a parameter $\theta_k$ that is specific to annotator $k$:

$$P(R_i, R_j|\theta_k) = \frac{1}{1 + e^{-\theta_k(R_i - R_j)}}, \qquad (4)$$

This new formulation has the following properties:

- **Maintain symmetry**: The symmetry to the model's abilities $R_i$ and $R_j$ is preserved even after modifying the constant $C$ to an annotator-related parameter $\theta_k$, such that $P(R_i, R_j|\theta_k) + P(R_j, R_i|\theta_k) = 1$

- **Discriminative ability ($\theta_k > 0$)**: When the abilities of two models are identical, the change in win probability caused by small variations in ability values is positively correlated with annotator's ability $\theta_k = 4\frac{\partial P(R_i,r)}{\partial R_i}\big|_{R_i=r}$. Therefore, the annotator's ability $\theta_k$ represents the maximum discriminative ability.

- **Anomalous annotator ($\theta_k < 0$)**: When the discriminative ability $\theta_k$, it is observed that for any model $i$ with greater ability than model $j$, annotator $k$ perceives the probability of model $i$ winning as less than 0.5. This indicates that it is an anomalous annotator.

To estimate the parameters of the probability function, we consider its logarithmic likelihood function similarly:

$$\sum_{(i,j,k,W_{ij})\in S} W_{ij} \ln P(R_i, R_j|\theta_k) + W_{ji} \ln P(R_j, R_i|\theta_k).$$
$$(5)$$

After modifying the probability function, we need to account for both the ELO scores of the models $R = (R_1, \ldots, R_N)$ and the abilities of the annotators $\Theta = (\theta_1, \ldots, \theta_M)$ during gradient descent. For a model $n \in [N]$ and annotator $m \in [M]$, the gradients of $\ln L$ to them can be expressed as:

$$\frac{\partial \ln L}{\partial R_n} = \sum_{(x,j,k,W_{nj})\in S} \theta_k(W_{nj} - P(R_n, R_j|\theta_k)),$$

$$\frac{\partial \ln L}{\partial \theta_m} = \sum_{(i,j,m,W_{ij})\in S} (R_i - R_j)(W_{ij} - P(R_i, R_j|\theta_m)).$$
$$(6)$$

This method allows us to simultaneously estimate the annotators' abilities during the MLE process. Beyond the concept of discrimination introduced by the improved probability function, we should also explore the practical significance of this ability estimation in the context of the arena. Through analysis, we find that the estimated annotator ability $\theta_k$ exhibits the following two properties:

**Theorem 4.2.** *Given that $\theta$ represents the ability of annotators estimated by am-ELO, the following conclusions can be drawn:*
*(1) If two annotators label the same set of samples $W_{ij}, W'_{ij}$ with abilities $\theta_1$ and $\theta_2$ ($\theta_2 > \theta_1$), then:*

$$\sum_{(i,j,W_{ij})\in S'} (R_i - R_j)W_{ij} < \sum_{(i,j,W'_{ij})\in S'} (R_i - R_j)W'_{ij}.$$

*(2) If $\theta_k < 0$, for each positive sample $(i, j, k, 1)$ of annotator $k$, its loss $\frac{\partial \ln l}{\partial R_i} < 0$, and for each negative sample $(i, j, k, 0)$ of annotator $k$, $\frac{\partial \ln l}{\partial R_i} > 0$.*

From Theorem 4.2, it is evident that the annotator abilities derived from MLE have practical significance. Specifically, $\sum_{(i,j,W_{ij})\in S}(R_i-R_j)W_{ij}$ can be interpreted as the correlation between the annotations $W_{ij}$ and the rankings $R_i - R_j$. Theorem 4.2 (1) implies that a higher annotator ability corresponds to a greater value of $\sum_{(i,j,W_{ij})\in S'}(R_i - R_j)W_{ij}$, meaning that *a larger $\theta_k$ indicates that the annotations from annotator k are more consistent with the overall rankings*. Meanwhile, Theorem 4.2 (2) suggests that an annotator with negative ability might annotate inconsistently or arbitrarily, and am-ELO can identify these anomalous annotators.

**Normalization** Although this method has strong interpretability for modeling annotators, it is not difficult to observe that, for such an optimization problem, if $(R_1^*, \cdots, R_N^*, \theta_1^*, \cdots, \theta_M^*)$ is an extreme point, then

---

**Algorithm 2** The am-ELO Rating System

   **Input:** Dataset $S$, Learning Rate $\alpha$, Epoch $Epoch$
   **Initialize:** Score of Models $R$ and annotators' ability $\Theta$
   **for** $t = 1$ **to** $Epoch$ **do**
      Calculate MLE: $\ln L \leftarrow \text{MLE}(R, \Theta, S)$
      Optimize: $R \leftarrow R + \alpha \frac{\partial \ln L}{\partial R}$, $\Theta \leftarrow \Theta + \alpha \frac{\partial \ln L}{\partial \Theta}$
      Normalization: $\Theta \leftarrow \frac{\Theta}{\mathbf{1}^T \cdot \Theta}$
   **end for**
   **Output:** ELO Score and annotators' ability $(R, \Theta)$.

---

**Algorithm 3** The Stable Arena framework

   **Input:** Learning Rate $\alpha$, Epoch $Epoch$, Ability Threshold $\epsilon$.
   **Initialize:** Dataset $S \leftarrow \emptyset$, Data quantity threshold $\delta$.
   **while** True **do**
      $S \leftarrow S \cup S_{new}$
      **for** $k = 1$ **to** $M$ **do**
         $\delta_k = |\{(i, j, x, W_{ij})|x = k\}|$
      **end for**
      $S' \leftarrow \{(i, j, k, W_{ij})|\delta_k > \delta\}$
      $(R, \Theta) \leftarrow$ am-ELO$(S', \alpha, Epoch)$
      $(R_1, \cdots, R_N) = R$
      **Output:** ELO Score $(R_1, \cdots, R_N)$
      $(\theta_1, \cdots, \theta_N) = \Theta$
      Filter annotators: $S \leftarrow \{(i, j, k, W_{ij})|\theta_k > \epsilon\}$
   **end while**

---

$(\alpha R_1^*, \cdots, \alpha R_N^*, \frac{1}{\alpha}\theta_1^*, \cdots, \frac{1}{\alpha}\theta_M^*)$ is also an extreme point. Thus, when $\alpha < 0$, the model score ranking will be completely reversed, leading to potential instability. To mitigate this issue, we impose a constraint on the annotator's ability:

$$\theta_1 + \theta_2 + \cdots + \theta_M = 1. \tag{7}$$

From Theorem 4.2 (2), we know that $\theta_k > 0$ corresponds to users who annotate normally. The significance of this normalization operation is essentially based on the assumption that the majority of annotators in the group are labeling responsibly (Nowak & Rüger, 2010). Based on this assumption, we determine whether the model rankings should follow the original order or be reversed.

### 4.3. Stable Arena Framework

Algorithm 2 presents the pseudo-code for the am-ELO algorithm. The am-ELO algorithm performs gradient descent (Ruder, 2016) on the negative log-likelihood function over the entire dataset to find the extreme point, ultimately returning both the model scores and annotator abilities. Specifically, when considering only the m-ELO algorithm, the concavity of its log-likelihood function enables the use of Newton's method (Galántai, 2000; Kelley, 2003) during optimization. This allows for dynamic adjustment of the learning rate, thereby improving convergence efficiency.

Building on the improvements to the ELO method discussed earlier, we introduce the Stable Arena Framework, a novel paradigm for arena-based evaluation, as detailed in Algorithm 3. To ensure more robust evaluations, we carefully screen the annotated data both before and after applying the am-ELO method. Specifically, upon incorporating new annotation samples, we first filter out annotators who have fewer than $\delta$ annotation records. This is crucial because annotators with fewer records tend to produce less reliable results. However, this does not imply permanent exclusion; once such annotators accumulate a sufficient number of annotations, their records will be reconsidered.

After evaluating both models and annotators, we further refine the process by filtering annotators based on their estimated abilities. Annotators with negative ability values, or those with ability values below a threshold $\epsilon$, are deemed detrimental to the evaluation process. For these annotators, we either issue warnings or exclude them entirely from further evaluations. Moreover, since a higher $\theta$ indicates greater consistency between the annotations and the overall ranking, the LLM evaluation platform can reward annotators proportionally to their demonstrated abilities.

## 5. Experiments

In this section, we introduce and compare the performance of our proposed method with the traditional ELO method in predicting annotation results, highlighting the superior modeling capability of am-ELO. Additionally, we demonstrate the limitations of the traditional ELO method through a comparison of model rankings produced by various ELO methods and a case study. Next, we validate the convergence of the ELO rankings generated by our method, further reinforcing the validity of our approach for evaluating LLMs. Finally, to assess the stability of the ELO method, we apply four different strategies to perturb the annotators. Our results show that our method not only maintains stability in the tests but also effectively identifies anomalous annotators, emphasizing the superiority of our approach.

### 5.1. Dataset

We conduct experiments on a real annotation dataset, **Chatbot** (Zheng et al., 2023), which was collected from 13,000 distinct IP addresses in the Chatbot Arena between April and June 2023. The dataset consists of 33,000 curated conversations with pairwise human preferences. Each entry includes a question ID, the names of two models, their full conversation transcripts, the annotator's vote, and its ID. Due to the requirement for MLE in this experiment, individual samples may introduce instability. Consequently, we excluded annotator samples with fewer than 50 annotated records. The statistical information of the filtered dataset is shown in Table 1.

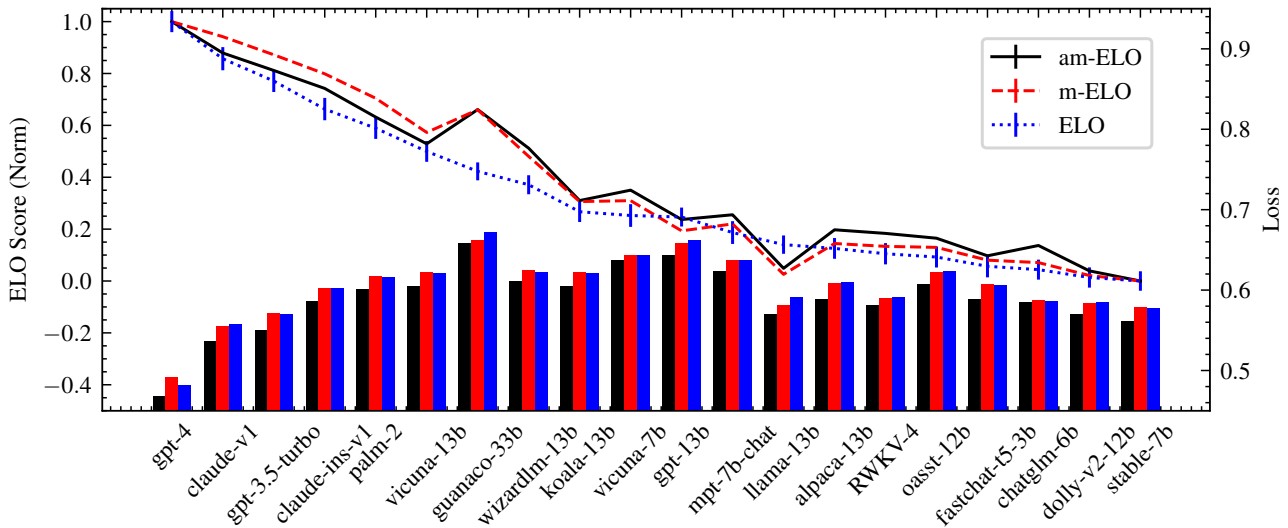

*Figure 3.* The result of each LLMs on different evaluation method. Specifically, the line chart represents the normalized ELO scores↑ (ranging from 0 to 1) of each LLM under different evaluation methods. The bar chart represents the Loss↓ (log-likelihood function) of each LLM's match records under different evaluation methods.

*Table 1.* Statistics of the dataset

| Dataset | **Chatbot** |
|---|---|
| #Annotators | 42 |
| #Models | 20 |
| #Response logs | 4321 |
| #Response logs per annotator | 102.88 |
| #Response logs per model | 216.05 |
| #Response logs per model pair | 22.74 |

*Table 2.* The Performance of ELO method for prediction.

| Method | **MSE↓** | **AUC↑** |
|---|---|---|
| ELO | 0.1238± 0.0031 | 0.7492± 0.0068 |
| m-ELO | 0.1234± 0.0029 | 0.7503± 0.0066 |
| am-ELO | **0.1208**± 0.0034 | **0.7581**± 0.0067 |

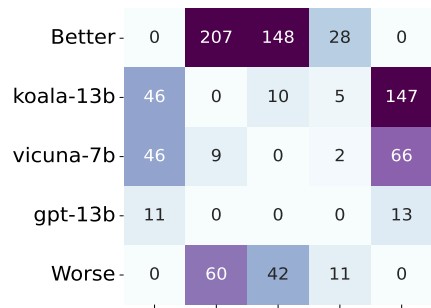

*Figure 4.* The heatmap shows the number of victories in battles between various models (Three models with similar abilities, koala-13b, vicuna-7b, gpt-13b, and the better or worse models than them). Each number in the figure represents the times the row model wins the column model in the battle.

## 5.2. Setting

In this experiment, we consider a baseline model, the traditional **ELO** method, alongside two methods we proposed: **m-ELO** and **am-ELO**. For the iterative ELO method, we perform repeated experiments by shuffling the dataset 1000 times and averaging the results. The MLE is solved using the gradient descent (GD) approach with a learning rate of 0.1 and a fixed number of 2000 iterations. The code can be found in the github: https://github.com/bigdata-ustc/am-ELO.

## 5.3. Result and Case Study

The bar chart in Figure 3 presents the mean log-likelihood loss for each method. As shown, the loss difference between m-ELO and ELO, which share the same probability function, is minimal, while the loss for am-ELO is significantly lower than the other two methods. This indicates that am-ELO demonstrates better fitting ability. Furthermore, as shown in Table 2, am-ELO significantly outperforms the other two baseline models in prediction tasks, suggesting that am-ELO exhibits superior generalization ability. This

also demonstrates that the improved probability function effectively models the annotators.

Meanwhile, the line chart in Figure 3 illustrates the ELO scores obtained from the three ELO methods. It is clear that the ranking trends of our proposed methods align closely with the traditional ELO method. However, there are some differences in the rankings of specific models, such as koala-13b, vicuna-7b, and gpt-13b.

To analyze these models with similar abilities, we categorize

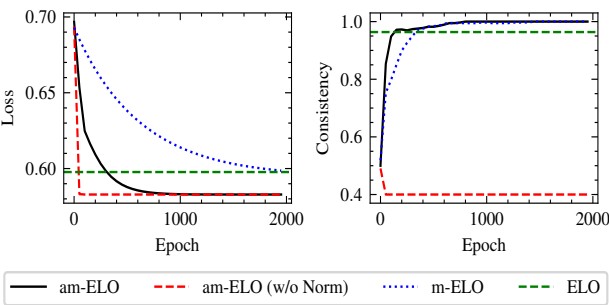

*Figure 5.* The Loss and Consistency of the evaluation method at each epoch on the Chatbot dataset.

the remaining models into two groups based on their ELO scores: "Better" and "Worse", representing models that are better or worse than the aforementioned models. We visualize the number of matches between these models. As shown in Figure 4, each number represents the number of times the model in the row defeated the model in the column. For example, the first row and third column indicate that vicuna-7b lost to "Better" models 148 times. From this, we observe that although the head-to-head records between koala-13b and vicuna-7b do not differentiate their abilities, both models defeated the same number of "Better" models. Meanwhile, vicuna-7b lost to fewer "Better" and "Worse" models. Based on this result, we conclude that vicuna-7b is stronger than koala-13b, which aligns with the rankings provided by both am-ELO and m-ELO.

However, due to Koala-13B's large number of victories over "Worse" models, the traditional ELO method disproportionately weighs these victories during the scoring process, ultimately ranking Koala-13B higher than Vicuna-7B. This issue suggests that avoiding strong opponents and repeatedly defeating weaker ones could artificially inflate a model's ranking, which is an undesirable outcome.

### 5.4. The Convergence and Efficiency of ELO Methods

In this subsection, we discuss the convergence and efficiency of the proposed am-ELO. Our comparison methods not only three mentioned model but also **am-ELO (w/o Norm)**, where normalization is not performed during training. To analyze the convergence and efficiency of the results obtained by each evaluation method, we record the loss (**Loss**) during the gradient descent process. Additionally, we perform five random initializations of the model parameters and calculate the consistency of the rankings (**Consistency**) (Hastie & Tibshirani, 1997) of the ELO scores output by these five models at each epoch. It should be noted that the iterative process of the traditional ELO method differs from the gradient descent approach of MLE. Therefore, we directly record the final output loss and consistency for the traditional ELO method. The results are shown in Figure 5.

As observed from the loss, the three MLE-based methods

all converge to a local minimum within a limited number of iterations. The loss at convergence for m-ELO is nearly identical to that of the traditional ELO, which is expected since both methods share the same probability estimation function. This once again demonstrates that m-ELO and traditional ELO are essentially equivalent. Moreover, am-ELO (w/o Norm) converges the fastest, followed by am-ELO, with m-ELO being the slowest. This is because am-ELO has more adjustable parameters compared to m-ELO, and am-ELO (w/o Norm) benefits from fewer constraints during the gradient descent process. However, as seen from the consistency, am-ELO (w/o Norm) quickly converges to different local minima, and its consistency stabilizes at 0.4. This suggests that the five outputs of this method exhibit two ordered sequences and three reversed sequences ($\frac{C_2^2 + C_3^2}{C_5^2} = 0.4$). On the other hand, am-ELO not only achieves stable rankings after sufficient gradient descent iterations but does so more efficiently than m-ELO. This demonstrates that the proposed am-ELO method strikes a balance between convergence and efficiency.

### 5.5. The Stability of Various ELO Methods

Since directly verifying the stability of the am-ELO method during the evaluation process is challenging, we use simulation experiments to introduce perturbation to the annotators. Specifically, we perturb the annotators' results using four strategies to simulate the presence of anomalous annotators that may occur during testing:

- **Random**: If model A wins, the result will have a 50% chance of being changed to "Tie" and a 50% chance of being changed to "model B wins", vice versa.

- **Equal**: All results are changed to "Tie".

- **Flip**: If model A wins, the result will be flipped to "model B wins", and vice versa. The outcome "Tie" remains unchanged.

- **Mixed**: A random selection is made from the first three perturbation strategies for each instance.

These perturbations mimic scenarios where intentional mislabeling occurs in annotations. Considering that the majority of annotators in the arena will annotate normally, the number of perturbations in our simulation experiment will not exceed half of the total number of annotators. We expect a stable scoring method to have two key properties: (1) it should produce ELO rankings consistent with those without perturbations, and (2) it should identify the anomalous annotators. The ground truth for the consistency of the ELO score is the pairwise comparison between the ELO rankings with and without perturbations, and the ground truth for identifying anomalous annotators is the F1-score

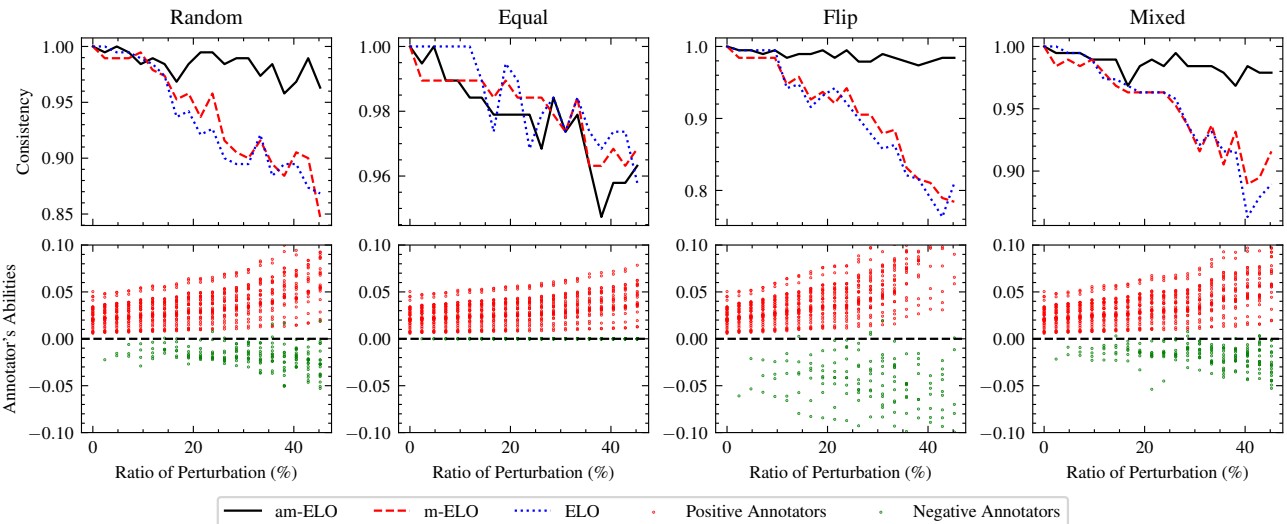

*Figure 6.* This figure contains four line charts and four scatter plots, corresponding to the ELO score consistency under the four types of perturbation, as well as the changes in annotator abilities obtained from am-ELO as the level of perturbation increases.

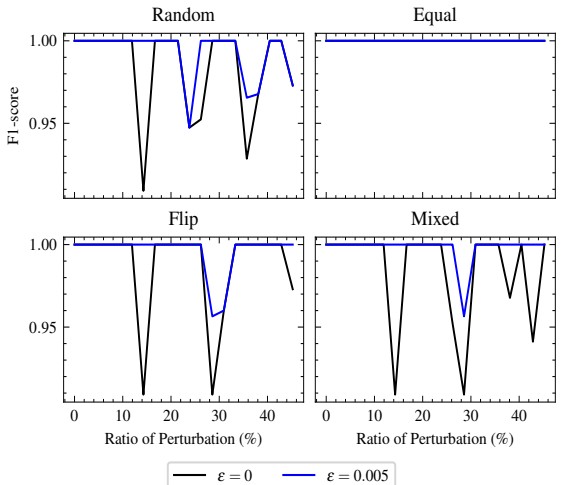

*Figure 7.* The line chart of F1-score for outlier detection at different thresholds under the four types of perturbation.

(Chen & Lin, 2006) of the annotators' abilities obtained from am-ELO. A higher F1-score indicates better accuracy in detecting the perturbations.

The line charts in Figure 6 show the relationship between the ratio of perturbations and the consistency of ELO scores. We observe that am-ELO maintains higher consistency across various types of perturbations. Specifically, aside from the fact that the "Equal" perturbation itself is unlikely to affect rankings, leading to high consistency across all ELO methods, in the other three perturbation scenarios, am-ELO reduces the inconsistency rate to 30% compared to m-ELO or traditional ELO. Meanwhile, the scatter plot at the bottom of Figure 6 shows the changes in annotator abilities under each perturbation. Red dots represent annotators who were normal, while green dots represent those who were anomalous. It is clear that nearly all anomalous annotators

have ability scores below 0, indicating they are identified as noise points. Additionally, Figure 7 presents the F1-scores for detecting perturbations under thresholds of 0 and 0.005. Under different perturbations, the recognition accuracy reached 90% when $\epsilon = 0$, and even up to 95% when $\epsilon = 0.005$. These results demonstrate that our method effectively detects perturbations, models the annotators, and maintains the consistency of results, thereby alleviating the problem of ELO inconsistency.

## 6. Conclusion

In this study, we explored the instability of the ELO method in the context of LLM evaluation, emphasizing its impact on the reliability of evaluation outcomes. To address this issue, we introduced the Stable Arena Framework, which utilizes the MLE approach for ELO rating estimation and incorporates annotator ability parameters into the probability function. Our experiments demonstrated that am-ELO not only achieves more stable convergence but also effectively identifies anomalous annotators, resulting in rankings that are more aligned with human intuition. These findings suggest that our approach can significantly reduce the instability of ELO, enhancing the credibility and robustness of LLM evaluation, while providing a more stable and easily implementable framework for arena-based evaluation.

However, our method has certain limitations. Specifically, the dimensions of annotator modeling are somewhat simplistic, as it primarily captures the annotator's discriminatory ability and consistency with other annotators. This makes it challenging to fully capture the annotator's broader capabilities. In future work, we aim to refine the design of annotator ability dimensions to better leverage crowdsourcing for arena-based evaluation.

## Acknowledgements

This research was supported by grants from the National Key Research and Development Program of China (Grant No. 2024YFC3308200), the National Natural Science Foundation of China (62337001), the Key Technologies R & D Program of Anhui Province (No. 202423k09020039), China Postdoctoral Science Foundation (Grant No. 2024M760725) and the Fundamental Research Funds for the Central Universities.

## Impact Statement

This paper presents work whose goal is to advance the field of Machine Learning. There are many potential societal consequences of our work, none which we feel must be specifically highlighted here.

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

## A. Proofs of Theorem 4.1

*Proof.* Assume $R_1 = 0$ and consider the remaining variables $(R_2, \cdots, R_N)$. For each sample $(i, j, W_{ij})$, consider the log-likelihood function $\ln l$ for this sample is given by:

$$\ln l = W_{ij} \ln P(R_i, R_j) + W_{ji} \ln P(R_j, R_i).$$

The second-order partial derivatives of $\ln l$ are:

$$\frac{\partial^2 \ln l}{\partial R_i^2} = -C^2 P(R_i, R_j)(1 - P(R_i, R_j)), i \neq 1,$$

$$\frac{\partial^2 \ln l}{\partial R_i \partial R_j} = C^2 P(R_i, R_j)(1 - P(R_i, R_j)), i, j \neq 1,$$

Now, let the number of matches between model $i$ and model $j$ be $\delta_{ij}$ and define $a_{ij} = \delta_{ij} C^2 P(R_i, R_j)(1 - P(R_i, R_j))$. For the Hessian matrix (Thacker, 1989) of the log-likelihood function $\frac{\partial^2 \ln L}{\partial \mathbf{R} \partial \mathbf{R}^T}$, its quadratic form (O'Meara, 2013) can be expressed as:

$$\mathbf{x} \frac{\partial^2 \ln L}{\partial \mathbf{R} \partial \mathbf{R}^T} \mathbf{x}^T = -\sum_{i=2}^{N} \sum_{j=2}^{N} a_{ij}(x_i - x_j)^2 - \sum_{i=2}^{N} a_{i1} x_i^2 - \sum_{j=2}^{N} a_{1j} x_j^2.$$

Note that $a_{ij} \geq 0$, therefore:

$$\mathbf{x} \frac{\partial^2 \ln L}{\partial \mathbf{R} \partial \mathbf{R}^T} \mathbf{x}^T \leq 0.$$

The equality holds if and only if $x_i = x_j = 0$, i.e. $\mathbf{x} = \mathbf{0}$. Since the quadratic form is strictly negative for all non-zero vectors $\mathbf{x}$, the Hessian matrix $\frac{\partial^2 \ln L}{\partial \mathbf{R} \partial \mathbf{R}^T}$ is negative definite (Johnson, 1970). This implies that the log-likelihood function $\ln L$ is concave. Therefore, $\ln L$ can have at most one extreme point (Boyd & Vandenberghe, 2004), ensuring the uniqueness of the maximum likelihood solution. $\square$

## B. Proofs of Theorem 4.2

*Proof.* (1) For annotators 1 and 2, the following formula can be obtained from Equation 6:

$$\frac{\partial \ln L}{\partial \theta_1} = \sum_{(i,j,W_{ij}) \in S'} (R_i - R_j)(W_{ij} - P(R_i, R_j|\theta_1))$$

$$\frac{\partial \ln L}{\partial \theta_2} = \sum_{(i,j,W'_{ij}) \in S'} (R_i - R_j)(W'_{ij} - P(R_i, R_j|\theta_2))$$

Since $\frac{\partial \ln L}{\partial \theta_1} = \frac{\partial \ln L}{\partial \theta_2} = 0$, the difference between the two equations can be obtained:

$$\sum_{(i,j,W_{ij}) \in S'} (R_i - R_j)(W_{ij} - W'_{ij}) = \sum_{(i,j,W'_{ij}) \in S'} (R_i - R_j)(P(R_i, R_j|\theta_1) - P(R_i, R_j|\theta_2))$$

According to the Lagrange mean value theorem (Shi-gu, 2014), the following derivation can be derived:

$$= \sum_{(i,j,W'_{ij}) \in S'} (R_i - R_j)^2 P_{ij}(\xi_{ij})(1 - P_{ij}(\xi_{ij}))(\theta_1 - \theta_2)$$

Due to $P_{ij}(\xi_{ij})(1 - P_{ij}(\xi_{ij})) > 0$ and $\theta_1 < \theta_2$:

$$\sum_{(i,j,W_{ij}) \in S'} (R_i - R_j)(W_{ij} - W'_{ij}) < 0$$

(2) Because of $0 < P(R_i, R_j|\theta_k) < 1$ and $\theta_k < 0$, for each positive sample $(i, j, k, 1)$ of annotator $k$, we have $\frac{\partial \ln l}{\partial R_i} = \theta_k(1 - P(R_i, R_j|\theta_k)) < 0$. Similarly, for each negative sample $(i, j, k, 0)$ of annotator $k$, we have $\frac{\partial \ln l}{\partial R_i} = \theta_k(0 - P(R_i, R_j|\theta_k)) > 0$. $\square$

