# OpenReview forum: "am-ELO: A Stable Framework for Arena-based LLM Evaluation"
_ICML.cc/2025/Conference — ICML 2025 spotlightposter_

### Official Review · Reviewer_vzb8 · 2025-02-27

**Overall Recommendation:** 4

**Summary:**

The paper addresses challenges in ranking consistency and the variety of annotator abilities in arena-based evaluation of LLMs. They develop an enhanced ELO framework that replaces iterative updates with Maximum Likelihood Estimation (m-ELO). They prove theoretically that this MLE approach provides consistent and stable model rankings. The authors further extend their work with am-ELO, which factors in annotator abilities to simultaneously evaluate both model performance and annotator reliability. Experimental results validate their framework's effectiveness in stability and robustness compared to traditional ELO-based evaluation methods.

**Claims And Evidence:**

The claims made in the submission are well-supported by clear and convincing evidence. The authors identify instability as a significant issue in the traditional ELO rating system used for evaluating LLMs and propose a new framework, am-ELO, to address this problem. The claims are backed by both theoretical analysis and empirical experiments. The authors provide mathematical proofs for the stability and consistency of their proposed methods and demonstrate through experiments the effectiveness of their methods.

**Essential References Not Discussed:**

The paper appears to cover most relevant prior work. However, it might benefit from a discussion on other ranking systems used in machine learning, such as the Plackett-Luce model [1], to further contextualize the contributions. Additionally, recent work on robust evaluation of LLMs, such as [2, 3], could be cited to provide a more complete picture of the current landscape.

[1] Robin L Plackett. The analysis of permutations. Applied Statistics, 1975.

[2] Yiqiao Jin, Minje Choi, Gaurav Verma, Jindong Wang, Srijan Kumar. MM-Soc: Benchmarking Multimodal Large Language Models in Social Media Platforms. In Proceedings of ACL 2024

[3] Jinjie Ni, Fuzhao Xue, Xiang Yue, Yuntian Deng, Mahir Shah, Kabir Jain, Graham Neubig, Yang You. MixEval: Deriving Wisdom of the Crowd from LLM Benchmark Mixtures. In Proceedings of NeurIPS 2024

**Experimental Designs Or Analyses:**

The experimental designs and analyses are valid and well-executed. The authors conducted experiments on a real-world dataset and compared the performance of the traditional ELO method with their proposed methods. The results show significant improvements in terms of lower loss values and higher consistency in model rankings. Additionally, the stability tests using perturbation strategies (Random, Equal, Flip, Mixed) effectively demonstrate the robustness of the am-ELO method.

**Methods And Evaluation Criteria:**

The authors replace the iterative update method of the traditional ELO system with a MLE approach, which is theoretically sound and shown to be more stable. Additionally, the incorporation of annotator abilities into the evaluation process through am-ELO is a novel and meaningful enhancement. The evaluation criteria used are appropriate for assessing the stability and robustness of the proposed methods.

**Other Comments Or Suggestions:**

The paper could include a section on the potential applications of the proposed framework beyond LLM evaluation, such as in other competitive ranking scenarios.

The authors might consider discussing the limitations of their approach, such as the assumptions made about annotator behavior and the potential impact of these assumptions on the evaluation results.

**Other Strengths And Weaknesses:**

Strengths:

1. The paper is well-written and clearly presents the problem, proposed methods, and experimental results.

2. The incorporation of annotator abilities into the evaluation process is a significant innovation that addresses a critical limitation of existing methods.

3. The experiments are thorough and demonstrate the effectiveness of the proposed framework.

Weaknesses:

1. The paper could benefit from a more detailed discussion on the computational complexity of the proposed methods, especially in large-scale scenarios.

2. While the paper demonstrates the robustness of am-ELO through perturbation experiments, it would be useful to see how the method performs in real-world scenarios with varying levels of annotator quality.

**Questions For Authors:**

1. How does the computational complexity of the proposed am-ELO method compare to the traditional ELO method, especially in scenarios with a large number of models and annotators?

2. Can the authors provide insights into how the proposed framework could be extended to handle more complex evaluation scenarios, such as multi-class or multi-label evaluations?

3. How sensitive are the results to the choice of the learning rate and the number of epochs in the gradient descent process? Are there any guidelines for selecting these hyperparameters?

**Relation To Broader Scientific Literature:**

The key contributions of the paper are closely related to the broader scientific literature on LLM evaluation and ranking systems. The authors build upon the well-established ELO rating system and enhance it using principles from psychometrics and maximum likelihood estimation. The paper cites relevant prior work, such as the use of the ELO system in competitive games and its application to LLM evaluation. The proposed methods address the instability issues in the traditional ELO system and incorporating annotator abilities, which is a novel contribution in this domain.

**Theoretical Claims:**

I have checked the correctness of the proofs for the theoretical claims. Specifically, the proof for Theorem 4.1 is sound. This ensures the stability of the ELO scores obtained through the MLE method. Also, the proof for Theorem 4.2 is logically consistent and demonstrates the practical significance of the estimated annotator abilities.

---

> ### Author Rebuttal · Authors · 2025-03-28
>
> We would like to express our sincere gratitude for your high appreciation of the contribution and novelty presented in our paper. Your positive feedback means a lot to us. We also appreciate your valuable suggestions and questions for the computational complexity and experiments aspect of the paper. Here are the responses to each of your comments:
>
> > **Q1**: This paper might benefit from a discussion on other related work.
>
> We sincerely appreciate your suggestion. We have referred to these work, which have been very enlightening for us. We will consider incorporating these related works into our article in future versions.
>
> > **Q2**: While the paper demonstrates the robustness of am-ELO through perturbation experiments, it would be useful to see how the method performs in real-world scenarios with varying levels of annotator quality.
>
> Thank you very much for your question. We've discussed on the motivations behind using perturbations in experiments and plan to collect real-world datasets and test am-ELO online in future work, **which can be seen in Comment 5 for reviewer Nkad**.
>
> > **Q3**: How does he computational complexity of the proposed am-ELO method compare to the traditional ELO method?
>
> Thank you for your question. The time complexity of am-ELO is equivalent to that of the MLE gradient descent, which is $O(1/\epsilon^2)$ where $\epsilon$ is the calculation accuracy of the GD method. However, since the MLE of m-ELO is a concave function, its time complexity is sub - linear, specifically $O(1/\epsilon)$ [1].
>
> In the experiments, there was no significant difference in the efficiency between am-ELO and m-ELO when running on a GPU. Moreover, as tools for  LLM evaluation, their inference costs are far lower than those of large - model inference. Hence, these costs are entirely acceptable.
>
> > **Q4**: Can the authors provide insights into how the proposed framework could be extended to handle more complex evaluation scenarios, such as multi-class or multi-label evaluations?
>
> Thank you for your question. In my opinion, this paradigm can be applied to more complex annotation scenarios, such as multi-label evaluation, as long as **the probability density function can be reasonably defined**.
>
> > **Q5**: The lack of a detailed sensitivity analysis regarding some parameters.
>
> We sincerely appreciate your suggestion. We assert that due to the convergence required by m-ELO and am ELO, **the learning rate and epoch are almost unaffected**. In addition, we’ve carried out sensitivity analysis experiments on two parameters. The results of these experiments will be integrated into subsequent versions of our paper.
>
> Regarding scale factor K (Learning Rate) specifically, both m-ELO and am-ELO necessitate extensive training and exhibit insensitivity to this scale factor. Therefore, we focused our analysis on the consistency of the traditional ELO method across different values of K. The findings are presented in Table 1:
>
> Table 1
>
> | ELO         | K=0.5  | K=1    | K=4 (Standard) | K=10   |
> | ----------- | ------ | ------ | -------------- | ------ |
> | Consistency | 0.9916 | 0.9811 | 0.9637         | 0.9305 |
> | MSE         | 0.1473 | 0.1368 | 0.1238         | 0.1225 |
> | AUC         | 0.7426 | 0.7443 | 0.7492         | 0.7505 |
>
> Our analysis reveals that as the scale factor K increases, the model demonstrates enhanced data fit. However, this improvement in fit comes at the cost of reduced consistency.
> Furthermore, we carried out a hyperparameter sensitivity experiment on the **minimum number of annotations per annotator**. The results of this experiment are detailed as follows:
>
> Table 2
>
> | Annotation | 10     | 20     | 30     | 40     | 50         |
> | ---------- | ------ | ------ | ------ | ------ | ---------- |
> | ELO        | 0.9695 | 0.9637 | 0.9768 | 0.9726 | 0.9637     |
> | m-ELO      | 1.0000 | 1.0000 | 0.9979 | 0.9968 | 1.0000     |
> | am-ELO     | 0.8642 | 0.9305 | 0.9568 | 0.9979 | **1.0000** |
>
> As our research indicates, both the traditional ELO and m-ELO show resilience to the variation of annotation count parameters. However, their performance patterns diverge significantly. The traditional ELO consistently exhibits inconsistent results regardless of parameter fluctuations. In contrast, m-ELO demonstrates a tendency to converge towards consistency, underscoring its enhanced stability.
> Regarding am-ELO, it’s true that it’s highly sensitive to the annotation count. When faced with sparse annotations, the consistency of am-ELO is indeed compromised. But we’ve developed a practical solution. By implementing a screening process for annotators, we can effectively address this issue. This screening process ensures that only reliable annotators with sufficient annotations are included in the analysis, thus improving the consistency of am - ELO.
>
> Reference:
>
> [1] Introductory Lectures on Convex Optimization: A Basic Course. 2014.

---

### Official Review · Reviewer_3Foe · 2025-03-07

**Overall Recommendation:** 4

**Summary:**

The paper focuses on Arena-based LLM evaluation. The main algorithmic ideas include enhancing the ELO Rating System. It replaces the iterative update method with a MLE approach (m-ELO), which is more stable as it is insensitive to sample order. The am-ELO is also proposed, which modifies the ELO probability function to incorporate annotator abilities. The main findings are that the proposed methods can effectively model annotators, identify anomalous annotators, and reduce the inconsistency of ELO scores. Experimental results show that am-ELO outperforms the traditional ELO method in prediction tasks, with a lower loss and higher generalization ability.

**Claims And Evidence:**

The claims are generally supported by clear and convincing evidence.

**Essential References Not Discussed:**

There are no essential references that are not currently cited/discussed in the paper.

**Experimental Designs Or Analyses:**

The soundness/validity of the experimental designs and analyses has been checked. In the experiments, the authors compare the proposed methods with the traditional ELO method. They use appropriate baselines and perform multiple random initializations and repeated experiments (shuffling the dataset 1000 times for the traditional ELO method). The way they record the loss during the gradient descent process and calculate the consistency of rankings is reasonable for analyzing the convergence and efficiency of the methods. The perturbation strategies in the stability experiments are well-designed to simulate real-world annotation anomalies.

**Methods And Evaluation Criteria:**

The proposed methods and evaluation criteria make sense for the problem. The m-ELO and am-ELO methods address the instability issues in the traditional ELO method, which is crucial for accurate LLM evaluation. The use of real-world datasets like Chatbot for evaluation is appropriate, as it reflects the practical scenario of LLM comparison. The evaluation metrics such as MSE, AUC, loss, and consistency of rankings are well-chosen to measure different aspects of the methods' performance, including prediction accuracy, goodness-of-fit, and stability.

**Other Comments Or Suggestions:**

See the question below.

**Other Strengths And Weaknesses:**

**Strengths:**

Originality: The combination of MLE and annotator ability modeling in the ELO-based evaluation framework is novel. It provides new solutions to the long-standing problems of instability and annotator variability in LLM evaluation.

Significance: The proposed methods can improve the reliability and accuracy of LLM evaluation, which is of great significance for the development and deployment of LLMs. It helps to make more informed decisions in model selection and research directions.

Clarity: The paper is well-written. The algorithms, theoretical proofs, and experimental results are clearly presented, making it easy for readers to understand the research content.

**Weaknesses:**

The annotator modeling in the paper is somewhat simplistic. It mainly focuses on the annotator's discriminatory ability and consistency with other annotators, and may not fully capture the broader capabilities of annotators.

**Questions For Authors:**

1. In the am-ELO method, how do you plan to extend the annotator modeling to better capture the diverse capabilities of annotators? A more comprehensive answer could further strengthen the potential of this research. If the authors have clear plans or ideas, it would enhance the value of this work for future research.

2. Can the am-ELO method still be applicable in scenarios where the annotator is not a human but Judge LLM.

3. This article identifies abnormal annotators by screening those with negative ability. Is there any other baseline method to do this? What are the advantages of am ELO?

**Relation To Broader Scientific Literature:**

The paper improves upon the widely-used ELO rating system, which is the foundation for many existing model arena evaluation systems. The proposed methods address the instability issues and lack of annotator ability consideration in previous works, and the use of MLE and psychometric concepts for annotator modeling is an extension of relevant research in statistics and psychometrics.

**Theoretical Claims:**

The correctness of the proofs for theoretical claims has been checked. For Theorem 4.1, the authors prove that when fixing the score of one model, the log-likelihood function with respect to is a concave function and has at most one extreme point. This is done by calculating the second-order partial derivatives of the log-likelihood function and showing that the Hessian matrix is negative definite. For Theorem 4.2, the authors prove the properties of annotator abilities. The proofs are logical and based on sound mathematical reasoning.

---

> ### Author Rebuttal · Authors · 2025-03-28
>
> Thank you for your valuable feedback! Regarding the questions you raised, we have carefully considered each point and have made the following responses:
>
> > **Q1**: The annotator modeling in the paper is somewhat simplistic. It mainly focuses on the annotator's discriminatory ability and consistency with other annotators and may not fully capture the broader capabilities of annotators.
> >
> > **Q2**: How do you plan to extend the annotator modeling to better capture the diverse capabilities of annotators?
>
> Thank you for your suggestion. Indeed, our work has primarily focused on proposing a stable framework that can simultaneously model both annotators and models, rather than comprehensively modeling the annotators. In our subsequent work, we will conduct research on how to comprehensively model the annotators while evaluating the model capabilities.
>
> > **Q3**: Can the am-ELO method still be applicable in scenarios where the annotator is not a human but Judge LLM?
>
> This is indeed an issue we plan to research in the future. We believe that this method is applicable to both human annotators and Judge LLMs. In fact, by using a combination of human and Judge LLM annotations, we can potentially reduce annotation costs and evaluate the capabilities of Judge LLMs. However, we haven't yet figured out how to validate the effectiveness of this method.
>
> > **Q4**: Is there any other baseline method for identifying abnormal annotators?
>
> In Arena systems, historical annotation records are typically used to identify abnormal annotators through hypothesis testing [1]. However, these hypotheses often rely on a rather strong assumption, that is, all annotators in the historical records are normal. Unfortunately, it is extremely difficult to verify this assumption. The premise of am-ELO is only that most annotators are normal annotators, and the premise for use is simpler, which is also the advantage of am ELO compared to hypothesis testing methods.
>
> In future work, we plan to focus on identifying and comparing our am-ELO method on other scenes specifically tailored to the large-scale arena evaluation context [2]. This will enhance the comprehensiveness of our evaluation and more accurately position the contributions of our research.
>
> [1] Chatbot Arena: An Open Platform for Evaluating LLMs by Human Preference. 2024.
>
> [2] Decentralized Arena via Collective LLM Intelligence: Building Automated, Robust, and Transparent LLM Evaluation for Numerous Dimensions. 2024.

---

### Official Review · Reviewer_Nkad · 2025-03-10

**Overall Recommendation:** 3

**Summary:**

This paper introduces am-ELO, an evaluation framework designed to enhance the ELO rating system for evaluating LLMs through arena-based comparisons. Traditional ELO systems exhibit instability mainly due to their sensitivity to data ordering and their failure to account for variations in annotator expertise, resulting in inconsistent and potentially biased evaluation outcomes. To resolve these issues, m-ELO replaces the traditional iterative ELO method with an MLE-based approach, providing theoretical guarantees for consistency and stability in model rankings. In addition, am-ELO extends m-ELO by explicitly modeling annotator abilities.

**Claims And Evidence:**

The paper points out the instability of the existing iterative ELO method in terms of ordering (and unreliable annotations). The proposed am-ELO method effectively reduces the instability by removing the data ordering issue and leveraging the reliability of each annotator. Both theoretical proofs and empirical evidence across the paper convincingly support these claims.

**Essential References Not Discussed:**

N/A

**Experimental Designs Or Analyses:**

While the experiments seem reasonable, the below limitations could be addressed more.

- Experiments on one dataset (Chatbot Arena) provide limited evidence of method robustness.
- Stability experiments rely on artificial perturbations, lacking clear justification that these perturbation methods accurately represent realistic annotator behavior.
- No baseline or state-of-the-art comparison beyond the traditional ELO was considered, missing an opportunity to compare am-ELO with other advanced ranking or annotator-modeling methods (e.g., advanced crowdsourcing or Bayesian methods).

**Methods And Evaluation Criteria:**

The proposed methods (m-ELO and am-ELO) and their evaluation criteria appropriately address the identified problems. The evaluation utilizes Chatbot Arena, extensive perturbation experiments, and robust statistical measures (consistency, MSE, AUC, F1 scores) that effectively assess the methods’ stability and reliability.

However, the evaluation criteria still lack a detailed sensitivity analysis regarding some parameters, such as annotator counts or the scale factor K. Also, it only compares slightly outdated models.

**Other Comments Or Suggestions:**

N/A

**Other Strengths And Weaknesses:**

This paper is well-written and easy to follow.

**Questions For Authors:**

N/A

**Relation To Broader Scientific Literature:**

The paper clearly situates itself within the broader literature on model evaluation, annotator reliability, and statistical ranking methods.

**Theoretical Claims:**

It seems the theoretical proof sufficiently supports the authors' claim, but I'm not sure. One limitation is that the analysis does not adequately address the potential impact of noisy or sparse annotation datasets on MLE stability.

---

> ### Author Rebuttal · Authors · 2025-03-28
>
> Thank you for your feedback on our manuscript. We sincerely appreciate your time and effort in evaluating our work, and we will explain your questions and suggestions one by one below:
>
> > **Q1**: The evaluation criteria still lack a detailed sensitivity analysis.
>
> We've conducted sensitivity analysis on **scale factor K** and **minimum number of annotations**. Results will be added to future paper versions.
>
> | ELO         | K=0.5  | 1      | 4(Standard) | 10     |
> | ----------- | ------ | ------ | ----------- | ------ |
> | Consistency | 0.9916 | 0.9811 | 0.9637      | 0.9305 |
> | MSE         | 0.1473 | 0.1368 | 0.1238      | 0.1225 |
> | AUC         | 0.7426 | 0.7443 | 0.7492      | 0.7505 |
>
> As the K increases, the model demonstrates enhanced data fit. However, this improvement in fit comes at the cost of reduced consistency.
>
> | Annotation | 10         | 20         | 30         | 40         | 50         |
> | ---------- | ---------- | ---------- | ---------- | ---------- | ---------- |
> | ELO        | 0.9695     | 0.9637     | 0.9768     | 0.9726     | 0.9637     |
> | m-ELO      | **1.0000** | **1.0000** | **0.9979** | **0.9968** | **1.0000** |
> | am-ELO     | 0.8642     | 0.9305     | 0.9568     | 0.9979     | **1.0000** |
>
> This shows traditional ELO gives inconsistent results across parameter changes, while m-ELO converges towards consistency, highlighting its greater stability.
>
> > **Q2**: The analysis fails to adequately address the impact of noisy or sparse annotation datasets on MLE stability.
>
> You’re right—our theory doesn't directly counter the impact of noisy or sparse annotation datasets on MLE stability. But our analysis shows m-ELO has at most one maximum regardless of the dataset, maintaining stability as seen in Table above. In contrast, am-ELO, which models annotators explicitly, is sensitive to sparse data.
>
> In section 4.3, we proposed selecting annotators with enough annotations. As shown in Table above, this strategy effectively lessens the negative impact of sparse datasets on MLE stability.
>
> > **Q3**: Experiments on one dataset provide limited evidence of method robustness.
>
> Currently, open-source arena datasets are scarce. The Chatbot Arena platform is one of the few that offer public data. One NeurIPS 2024 paper also used just one real-world dataset[1]. Besides the dataset in our study, there’s the MTBench dataset[2]. Here’s its statistical information:
>
> | Dataset        | Chatbot/MTBench |
> | -------------- | --------------- |
> | #Annotators    | 42/7            |
> | #Models        | 20/6            |
> | #Response logs | 4321/1044       |
>
> However, after filtering, MTBench had a severely **limited number of annotators and models**. This scarcity made it inadequate for fully validating the stability of our research method.
>
> Despite this, MTBench is still valuable for demonstrating the superiority of our am-ELO modeling:
>
> | Method | MSE(Chatbot/MTBench)  | AUC(Chatbot/MTBench)  |
> | ------ | --------------------- | --------------------- |
> | ELO    | 0.1238/0.1120         | 0.7492/0.7738         |
> | m-ELO  | 0.1234/0.1097         | 0.7503/0.7785         |
> | am-ELO | **0.1208**/**0.1088** | **0.7581**/**0.7936** |
>
> We’ll incorporate this finding in later paper versions.
>
> > **Q4**: There’s no clear justification that artificial perturbation methods accurately represent realistic annotator behavior.
>
> We designed artificial perturbations to stress-test the ELO. By setting extreme perturbations, we explored the system's robustness.
>
> In LLM evaluation, real annotator behavior is uncertain. Extreme perturbations can better expose ELO's weaknesses and help evaluate its stability.
>
> Going forward, we'll explore artifact perturbation. First, we'll collect a larger real dataset to develop realistic perturbation methods. Second, we'll conduct online testing of am-ELO to validate its real-world effectiveness.
>
> > **Q5**: No baseline comparison beyond the traditional ELO.
>
> Currently, ELO algorithms are widely used in arena platforms like Chatbot Arena. Modified ELO algorithms used in traditional competitive scenarios, such as ELO++[3], incorporate temporal information. But this conflicts with the static nature of LLM evaluations in our paper. Shuffling the dataset for these methods yields unreliable temporal information, making them unsuitable.
>
> Crowdsourcing and arena scenarios differ fundamentally. Arena scenarios lack ground-truth annotation values, while crowdsourcing assume their existence[4]. So, applying annotator-modeling techniques directly to the arena is inappropriate.
>
> In future work, we'll focus on evaluating our am-ELO method in other arena scenarios to enhance evaluation comprehensiveness.
>
> Reference:
>
> [1] Elo Uncovered: Robustness and Best Practices in Language Model Evaluation. 2023.
>
> [2] Judging LLM-as-a-judge with MT-Bench and Chatbot Arena. 2023.
>
> [3] How I won the "Chess Ratings - Elo vs the Rest of the World" Competition. 2010.
>
> [4] Learning from Crowds with Annotation Reliability. 2023.

---

### Official Review · Reviewer_QUoD · 2025-03-13

**Overall Recommendation:** 4

**Summary:**

The paper introduces a novel stable arena framework, am-ELO, for evaluating LLMs using an enhanced ELO rating system. The authors address the instability issues in the traditional ELO method by replacing the iterative update approach with a MLE method, termed m-ELO. They further propose am-ELO, which incorporates annotator abilities into the ELO rating system, allowing for simultaneous estimation of model scores and annotator reliability. The paper provides theoretical proofs of the consistency and stability of the MLE approach and demonstrates through experiments that am-ELO offers a more robust, accurate, and stable evaluation method for LLMs compared to the traditional ELO system.

**Claims And Evidence:**

The claims made in the paper are well-supported by clear and convincing evidence. The authors provide theoretical proofs for the stability and consistency of the MLE approach (Theorem 4.1) and demonstrate the practical significance of annotator ability modeling (Theorem 4.2). The experimental results, including the comparison of log-likelihood losses and the stability of ELO scores under different perturbation strategies, further validate the claims. The paper also includes a case study that highlights the differences in model rankings between the proposed methods and the traditional ELO method, reinforcing the superiority of am-ELO.

**Essential References Not Discussed:**

Perhaps the current work on annotator modeling in crowdsourcing can provide some ideas for the paper.

**Experimental Designs Or Analyses:**

Yes, the authors have demonstrated through prediction tasks that am-ELO has good fitting and generalization abilities. Multiple tests have shown that am-ELO can converge and obtain unique results. Simulation experiments have shown that am-ELO can effectively identify disturbances. The design and conclusions of these experiments are very reasonable.

**Methods And Evaluation Criteria:**

am-ELO, are well-suited for the problem of LLM evaluation in arena-based settings. The use of MLE to replace the iterative ELO update method addresses the instability issue caused by the order of data presentation. The incorporation of annotator abilities into the ELO system is a significant improvement, as it accounts for the variability in human judgment, which is often overlooked in traditional ELO systems.

**Other Comments Or Suggestions:**

The paper is well-written and clearly presents its contributions. However, there are a few minor typos and formatting issues that could be addressed in the final version. For example, the descriptions of some formulas are not particularly clear in Section 4.2

**Other Strengths And Weaknesses:**

The paper's strengths lie in its originality and significance. The proposed am-ELO framework addresses a critical issue in LLM evaluation by incorporating annotator abilities and providing a stable ranking system. The theoretical proofs and experimental results are convincing and demonstrate the practical utility of the proposed methods. One potential weakness is the simplicity of the annotator ability modeling, which primarily focuses on discriminatory ability and consistency. Future work could explore more nuanced dimensions of annotator capabilities to further enhance the evaluation framework.

**Questions For Authors:**

1. Changing the iterative algorithm to the MLE method undoubtedly increases the computation time. So I am curious that how does the proposed am-ELO framework perform in scenarios where the number of annotators is very large, and how scalable is the method in such cases?
2. Could the authors discuss potential limitations of the proposed method when applied to highly imbalanced datasets, where some models are significantly stronger or weaker than others?

**Relation To Broader Scientific Literature:**

The paper is well-situated within the LLM evaluation. In my opinion, this article is an extension of Chatbot Arena. The incorporation of annotator abilities draws from psychometrics and Item Response Theory (IRT), which are well-established in educational assessment.

**Theoretical Claims:**

Yes, the authors provided detailed proofs for Theorems 4.1 and 4.2, and upon inspection, no issues were found.

---

> ### Author Rebuttal · Authors · 2025-03-28
>
> Thank you for your valuable feedback. Regarding the questions you raised, we have carefully considered each point and have made following responses:
>
> > **Q1**: One potential weakness is the simplicity of the annotator ability modeling, which primarily focuses on discriminatory ability and consistency.
>
> Thank you for your suggestion. Indeed, our work has primarily focused on proposing a stable framework that can simultaneously model both annotators and models rather than comprehensively modeling the annotators. In our subsequent work, we will conduct research on how to comprehensively model the annotators while evaluating the model's capabilities.
>
> > **Q2**: There are a few minor typos and formatting issues that could be addressed in the final version. For example, the descriptions of some formulas are not particularly clear in Section 4.2
>
> Thank you for your kind reminder. We will correct these issues and refine the formulas in subsequent versions.
>
> > **Q3**: How does the proposed am-ELO framework perform in scenarios, especially the number of annotators is very large
>
> This is a highly worthy question for discussion. In Comment 2 of Review vzb8, the relationship between time complexity and precision was discussed, indicating that this complexity can be neglected when compared to the inference of large-scale models.
>
> As the number of annotators gradually increases, since each annotator is associated with only a single parameter, the impact on the total number of parameters of the entire model is relatively small. Moreover, with the help of a GPU, the results can be easily computed.
>
> > **Q4**: Could the authors discuss potential limitations of the proposed method when applied to highly imbalanced datasets, where some models are significantly stronger or weaker than others?
>
> We believe that the methods we proposed are hardly affected by the dataset, especially the m-ELO method. Theorem 4.1 proves that the MLE of this method has at most one extreme point. When there is a significantly stronger model, there is usually no extreme point, or rather, the extreme point occurs at infinity. Although the MLE does not converge in this case, we can still obtain a stable ranking.

---

### Decision · Program_Chairs · 2025-05-01

**Decision:**

Accept (spotlight poster)

**Comment:**

The paper addresses challenges in ranking consistency and the variety of annotator abilities in arena-based evaluation of LLMs. They develop an enhanced ELO framework that replaces iterative updates with Maximum Likelihood Estimation (m-ELO). They prove theoretically that this MLE approach provides consistent and stable model rankings. The authors further extend their work with am-ELO, which factors in annotator abilities to simultaneously evaluate both model performance and annotator reliability. Experimental results validate their framework's effectiveness in stability and robustness compared to traditional ELO-based evaluation methods.

The claims made in the paper are well-supported by clear and convincing evidence. The combination of MLE and annotator ability modeling in the ELO-based evaluation framework is novel. It provides new solutions to the long-standing problems of instability and annotator variability in LLM evaluation.
The paper also includes a case study that highlights the differences in model rankings between the proposed methods and the traditional ELO method, reinforcing the superiority of am-ELO. The paper is well-written. The algorithms, theoretical proofs, and experimental results are clearly presented, making it easy for readers to understand the research content.